# Co-Creation Approach with Action-Oriented Research Methods to Strengthen “Krachtvoer”; A School-Based Programme to Enhance Healthy Nutrition in Adolescents

**DOI:** 10.3390/ijerph18157866

**Published:** 2021-07-25

**Authors:** Marion D. Driessen-Willems, Nina H. M. Bartelink, Kathelijne M. H. H. Bessems, Stef P. J. Kremers, Conny Kintzen, Patricia van Assema

**Affiliations:** 1Department of Health Promotion, School of Nutrition and Translational Research in Metabolism (NUTRIM), Maastricht University, P.O. Box 616, 6200 MD Maastricht, The Netherlands; n.bartelink@maastrichtuniversity.nl (N.H.M.B.); k.bessems@maastrichtuniversity.nl (K.M.H.H.B.); s.kremers@maastrichtuniversity.nl (S.P.J.K.); p.vanassema@maastrichtuniversity.nl (P.v.A.); 2CITAVERDE College, 6049 CD Herten, The Netherlands; c.kintzen@citaverde.nl

**Keywords:** action research, co-creation, nutrition educational intervention, school support programme, adolescents, school health promotion, complex adaptive systems, sustainability

## Abstract

In recent years, the nutritional pattern of the Dutch adolescent has cautiously improved. However, progress can be gained if more Dutch adolescents adhere to the nutritional guidelines. School-based initiatives offer opportunities to deal with the unhealthy eating behaviours of adolescents via nutrition educational interventions. In designing and/or re-designing school-based interventions, it is important to enhance optimal context-oriented implementation adaptation by involving the complex adaptive school system. This paper elaborates on the way of dealing with the dynamic implementation context of the educational programme “Krachtvoer” (ENG: “Power food”) for prevocational schools, how the programme can be adapted to each unique implementation context, and how the programme can be progressively kept up to date. Following a co-creation-guided approach with various intersectoral stakeholders within and outside the school setting, action-oriented mixed research methods (i.e., observations, semi-structured interviews, focus group interviews, programme usage monitoring, and questionnaires) constantly provide input to develop the programme and its implementation strategy via continuous micro-process cycles. Successful co-creation of school-based health promotion seems to be dependent on proper intersectoral cooperation between research and practice communities, a national partner network that can provide project-relevant insights and establish capacity building aimed at improving contextual fit, and a time-investment balance in and between sectors.

## 1. Introduction

Unhealthy eating patterns among young people are considered a public health priority in many countries [1], including the Netherlands [2,3]. Consuming insufficient amounts of fruit and vegetables, too many high-energy snacks and sugary drinks, and inadequate mealtime intake leads to substantial health problems such as overweight and non-communicable diseases including coronary heart disease, diabetes, and different types of cancer [4]. These unhealthy eating patterns develop at a young age and may become worse when children get older and go from primary school to secondary school [5,6,7]. This prevalence pattern is also found in the Netherlands, where nutritional patterns have cautiously improved, but most adolescents (12–20 years old) do not adhere to the prevailing nutritional guidelines [7]. The results of the Dutch National Food Consumption Survey showed, for example, that adolescents consume on average less than half the recommended fruit intake of two pieces per day, consume many more high-energy snacks and drinks compared to other age groups, and the nutritional quality of adolescents’ breakfast and lunch is low [7]. These eating habits are even worse in adolescents following a lower education track [8,9,10]. School-based initiatives offer opportunities to deal with the problem of unhealthy eating among adolescents. Over the years, multiple school-based healthy diet promotion interventions have been developed, and the positive results of several of those interventions have indicated that this type of intervention can help improve eating habits [11,12]. Knowledge development in recent decades has shown that the conditions for the effectiveness of such interventions include theory and evidence-based intervention development, monitoring, and target group engagement [13]. 

It is increasingly being recognized that when developing and implementing such interventions, the complex dynamic context in which the interventions are implemented needs to be taken into account [14,15,16]. Schools can be considered complex adaptive systems where many actors and factors continuously interact [11,17]. The context is also constantly changing due to the open connection of the school with other systems (e.g., home, neighbourhood) and the changes that take place in those systems [18,19]. Consequently, each school has its own unique context. A healthy diet promotion intervention can be seen as one of the components of the complex adaptive system that continuously interacts with other components in the school context. This requires that an intervention and its implementation strategy should ideally be adapted to the specific and changing context within and outside the school at all times [20], implying a continuous effort from all involved parties. This requires developers of interventions and implementation strategies to shift from more traditional project-based working to programmatic working. Furthermore, adaptation to a context requires even higher levels of engagement of the target group and other actors within and outside the school in intervention development and implementation [20]. A co-creation approach seems essential in the development and implementation of health promotion interventions that are aimed at an optimal context-specific fit.

Co-creation has been generally defined as “the collaborative generation of knowledge by academics working alongside stakeholders from other sectors” [21]. In other words, in the co-creation process, new knowledge is generated via the honest, democratic, and meaningful engagement of key cross-sectoral stakeholders, making it sensitive to context and more likely to be implemented [22]. Co-creation is a relatively new concept that has been applied in many areas with diverse topics on which consensus on the domain-specific description is yet to be found [22,23]. The process of co-creation is also closely related to concepts such as ‘Utilization-focused Developmental Evaluation’ [15], ‘Coevolution’ [24], ‘Co-production’ [25], and ‘Process use’ [26]. Action-oriented research methods can facilitate the process of co-creation in a scientifically sound manner [27].

This paper describes a co-creation approach with action-oriented research methods as an engine to strengthen the Dutch nutrition educational programme “Krachtvoer” (ENG: “Power food”). Krachtvoer is a systematically developed and effective school-based educational intervention focusing on healthy nutrition for 12–14-year-old prevocational students [28,29]. It is complemented by a school support programme to facilitate optimal and sustainable implementation. Central themes are the consumption of fruit, snacks, and drinks, as well as breakfast and lunch intake. The first edition of Krachtvoer dates back to 2002, filling the need for an effective nutrition educational programme specifically for students in prevocational schools. [30]. The aim of this article is to elaborate on: (i) the way of dealing with the dynamic implementation context by using a co-creation approach, (ii) how the programme can be adapted to each unique implementation context, and (iii) how we are able to keep the programme, consisting of the educational intervention and school support programme, up to date.

## 2. Methods

### 2.1. The Krachtvoer Programme

Krachtvoer is an educational programme accredited by the National Institute for Public Health and Environment [31]. The Krachtvoer programme consists of two elements: (i) the educational intervention and (ii) the school support programme. The educational intervention is taught by teachers and targets first- and second-year students (aged 12–14) who are in prevocational schools, which host a majority of the total Dutch student population, and who have unhealthier eating habits compared to the total Dutch student population [9,10]. The educational intervention includes teacher manuals, lesson plans, worksheets, and ready-to-use Digi board lessons, with short videos, interactive quizzes, and instructions. The intervention consists of a minimum of eight lessons about healthy nutrition [32] and focuses on recommended behaviours related to fruit, snacks, and drinks, as well as intake at breakfast and lunch [33,34]. Regarding breakfast and lunch, these recommended behaviours come down to diet choices according to the Wheel of Five designed by the Dutch Nutrition Centre that includes the food categories of (1) vegetables and fruit; (2) spreadable and cooking fats; (3) fish, meat, egg, nuts, seeds, milk, and dairy products; (4) bread, cereal products, and potatoes; and (5) drinks [35]. These behaviours have been chosen because of their potential health benefits [36] and because adolescents can influence these behaviours themselves for the most part [37]. The educational intervention is complemented by a school support programme aimed at enhancing optimal context-oriented implementation adaptation. The school support programme provides teachers and school supporting staff, for example, with testimonials of active users, reference to training opportunities, guidance on dealing with individual students with specific conditions, motivation for and ready-to-use context-specific adaptation options of programme elements, and programme expansions within the nutritional theme and other themes of the Dutch Healthy School Programme.

During the initial development of Krachtvoer, intervention mapping principles [38] were used, including the use of available theoretical and empirical knowledge and additional data collection among students, teachers, and parents [30,39]. The theoretical backbone of the Krachtvoer intervention comprises theories on important elements of the behavioural change process, determinants of human behaviour, and self-management [40,41,42,43,44,45,46,47,48]. It acknowledges attention and enthusiasm for the topic and awareness of own behaviour as first prerequisites for change. Next comes the need to identify and improve behavioural determinants that act as barriers for change by members of the target population themselves. This is followed by creating personal goals and action plans as conditions for actual behavioural change and maintenance. Krachtvoer focuses on those potential personal and environmental determinants of dietary behaviours that have been identified as key determinants among youngsters and that can reasonably be expected to be influenced by an educational intervention, namely taste, awareness, self-efficacy, knowledge, attitudes, and social influences (subjective norms and exemplary behaviour) [49]. Change methods include active learning, self-re-evaluation, discussion, reinforcement, guided practice, demonstration, and repeated exposure [50]. Similarly, the theoretical backbone of the school support programme is based on relevant determinants of implementation (e.g., relative advantage, compatibility, and clear end-user guidelines), theory-based methods, and practical applications (e.g., testimonials, instructions, and videos) [32,51]. 

Following up on recommendations of previous evaluation research on the programme [30,39] while keeping the theoretical backbone intact, two key starting points for the co-creation approach were formulated: (i) to convert the outdated paper programme into a digital programme offered via a website that can easily be kept up-to-date, and (ii) to incorporate the function of Krachtvoer to serve as a catalyst for stimulating schools to take on or evolve their health promoting school approach [12,52]. This catalyst role was designed to stimulate a healthy lifestyle among students [12], and it also facilitates transfer-oriented learning [12,53] in which knowledge and skills learned in one context (e.g., creating an action plan with respect to nutrition) are applied to another topic (e.g., creating an action plan with respect to physical activity).

Derived from the self-management theory [40], Krachtvoer consists of four phases. Phase A aims to raise student attention and enthusiasm for the intervention and to introduce and prepare the students for the intervention’s learning methods. Phase B aims to develop awareness of own (risk) behaviour. Students learn about the nutritional guidelines, the health benefits of the recommended behaviours, their own behaviour, and whether their behaviour complies with the guidelines. Students are challenged to identify their own reasons for not complying with the guidelines. In phase C, the teacher offers students a choice of intervention components. Examples of intervention components are a product-tasting assignment, the preparation of a healthy recipe, a photo assignment to monitor own nutrition intake, and mapping the school environment regarding how well health behaviour is stimulated. Each intervention component in phase C is aimed at dealing with a specific type of barrier for behavioural change (e.g., taste, awareness, or self-efficacy). The teacher is encouraged to work on the most relevant identified barrier in a class. Phase D aims at sustainable behavioural changes by students. Students work on action plans, are challenged to convert the plans into actions, and evaluate—and modify if needed —their attempts. 

Previous evaluation studies have shown the short-term and longer-term effectiveness of Krachtvoer in enhancing targeted nutrition behaviours [30,39]. The previous studies also revealed that, although the intervention was implemented to a high degree of completeness and well-appreciated by students and teachers, improvements to the intervention and its implementation strategy are warranted [54]. Learning from previous research concerning intervention implementation [32], the educational intervention and the school support programme are sensitive to implementation- and context-related determinants [52,55,56,57].

### 2.2. Design of the Co-Creation Approach

The co-creation approach is a joint initiative of health promotion experts from Maastricht University and professionals from the educational organisation CITAVERDE. CITAVERDE College offers all levels of vocational education at four school locations in the Dutch province of South Limburg. Sustainability is the starting point of their educational vision: from healthy lifestyle to circular economy. The organization is specifically committed to the development of online learning tools. Together, these actors form the production team. In addition to the intensive daily collaboration between the production team members, the full team meets quarterly during the project period. The Maastricht University members of the production team substantiate their input with the most recent literature. Prevocational education locations of CITAVERDE act as primary collaboration partners in the co-creation approach, including the direct connection with students, teacher, parents, and school supporting staff. At the national level, the co-creation approach is supported by a group of partner organisations, many of which have a long history of involvement with the programme and represent expertise in the sectors of nutrition (Dutch Nutrition Centre), youth and parents (Dutch Youth Institute), physical education (Fontys University of Applied Sciences, School of Sport Studies), implementation (TNO Netherlands Organization of Applied Scientific Research), and the health promoting school approach (Schools for Health Consultancy). In addition, some of the developers of previous versions of the programme are involved via their current organisation (ResCon, Maastricht University). The partner organisations are continuously available for support requests. All stakeholders, including the production team members and the partner organisations, meet twice a year in a so-called partner meeting. The co-creation approach is financed by the Netherlands Organisation for Health Research and Development and augmented by financial contributions from the organisations represented in the production team. The budget includes financial compensation for the input of the school locations and the partner organisations. 

### 2.3. Conceptual Framework of Contextual Fit 

Action-oriented research is an essential component of the co-creation approach. The conceptual framework that guides the research is represented in Figure 1. The framework builds on previous work of Damschroder et al. [55], Fleuren et al. [57], and Schaap et al. [58], and it includes the Krachtvoer programme, the implementation process, and the context as its three main elements. The left frame visualises the Krachtvoer programme including the educational intervention and school support programme, both of which are inclusive of technology aspects (e.g., website development, Digi board lessons, and interactive quizzes), lay-out aspects (e.g., house style look, house style feel, and end-user experience-oriented design), and content aspects (e.g., teacher manuals, educational materials, and text on the website and in videos) [59,60]. The middle frame visualises the implementation process of the Krachtvoer programme. Indicators of successful implementation can be found in the balance between fidelity and adaptation [61], driven by contextual factors. Fidelity is an umbrella term for the degree to which an intervention was implemented as intended by the developers [58,62,63], predicting the intervention’s intended outcomes [61]. Adaptation indicates a bidirectional process in which a proposed change is modified to the needs, interests, and opportunities of the setting in which the intervention is implemented [58,63], providing guidance on how to optimise an intervention in a particular setting [61]. Five implementation indicators are distinguished: (i) dose (i.e., the amount of exposure to intervention components that is received by participants), (ii) adherence (i.e., the extent to which the intervention components are conducted and delivered according to the guidelines), (iii) quality of intervention delivery (i.e., how teachers deliver the intervention components), (iv) students’ responsiveness (i.e., the extent to which the students are engaged with the intervention), and (v) programme differentiation (i.e., the identification of essential intervention components for effective outcomes) [58]. The right frame visualises contextual aspects within the school, overlapping the inner and broader school setting [55,57]. It includes the characteristics of teachers (e.g., interest, affiliations, and classroom management), characteristics of students (e.g., health behaviour, social environment, and education level), school aspects (e.g., school size, facilities and location, prevailing health promotion elements, and climate), and contextual aspects in the broader school setting (e.g., developments on nutritional guidelines, the health promoting school approach, and curriculum renewing).

The framework’s rationale is that the implementation of health promotion programmes is determined by the programme itself, the continuously changing inner and broader school contexts, and the interaction between programme- and context-related aspects, i.e., contextual fit. Programme- and context-specific insights in these processes can therefore feed programme improvements that can improve implementation regardless of whether or not these are due to improved contextual fit or even capacity building in the implementation context [64]. The leading questions for the action-oriented research that was derived from the framework were therefore: how is the Krachtvoer programme implemented in prevocational schools and which programme-related, context-related, and interplay of programme- and context-related aspects influence implementation?

### 2.4. Phases in the Co-Creation Approach

The co-creation approach consists of three main phases relating to the programme development state. The approach is driven by the continuous micro-processes [65,66] of implementing, measuring, evaluating, and adapting the programme. These micro-process cycles have been adapted to the programme development state in terms of aim and timeframe, resulting in a different focus for each phase. In the first phase (school year of 2017–2018), programme components were separately developed in co-creation with one CITAVERDE location including one teacher and 18 classes with a total of 252 students. The existing paper programme version was redesigned to produce a draft version of the programme website including all revised programme components. In the second phase (school year of 2018–2019), the online draft version of the programme was implemented in co-creation with all four CITAVERDE schools, including 25 teachers who were not involved up till then, 32 classes with a total of 635 students, and 16 members of the school supporting staff. To finalise phase two, the programme website was published for nationwide use and a 10-year plan for sustainable programme use was created and committed to by the production team and partner network. The third phase (2019 onwards) is guided by a plan for sustainable programme use that serves the common goal: ensuring that Krachtvoer can be used, is used, and continues to be used at all prevocational schools in the Netherlands. Though this is a continuous and ongoing effort, in school year of 2019–2020, a start was made to work on the plan’s three main pillars: (i) website hosting, (ii) programme quality improvement, and (iii) programme dissemination strategy. In all programme development phases, mixed data collection methods among the various involved stakeholders continuously provide complementary input to further develop the programme (Table 1). The researchers in the production team collect, process, and analyse all data. Data collection and processing include interview guides, observation forms, and audio recordings. Qualitative data coding is performed in line with the conceptual framework and analysed using NVivo (QSR International Pty Ltd., Melbourne, Australia).

## 3. Discussion

Given the relatively new way of performing action-oriented research in the co-creation of school-based health promotion, it is important to consider how this approach facilitates optimal context-oriented implementation adaptation. The rich history of effect- and dissemination research concerning Krachtvoer is the basis underlying this action-oriented research [30,40]. Building on the core principles of the existing programme, incorporating end-user needs and enabling contextual fit reinforces the programme development and implementation process. Using an action-oriented research approach with continuous micro-process cycles facilitates the sustainable [67,68,69,70] quality improvement of the programme by increasing its adaptability to multiple and continuously changing contexts. In addition, to facilitate optimal context-oriented implementation adaptation, Krachtvoer has been created in collaboration with various types of stakeholders (e.g., students, teachers, school supporting staff, parents, and partner network).

A number of success factors and challenges are expected to be encountered in this co-creation approach. Collaboration between the research and school sectors is essential in co-creation due not just to complementary content (e.g., education methods and intervention content) but also to actual access to the school setting and the ability to test the programme with end-users. Collaboration with a national partner network is also essential. This can provide project-relevant insights and establish capacity building [64] concerning developments in the broader context (e.g., revised nutritional guidelines, curriculum renewing, and contemporary interests and needs of adolescents) resulting in increased contextual fit. Therefore, successful co-creation is primarily dependent on good intersectoral cooperation and mutual adaptation [15] between the key stakeholders of project parties. Co-creation needs to be an honest, two-way, and ongoing process in order to be a success [22]. Sufficient time for participating parties needs to be allocated for the co-creating approach. Taking up a co-creation process must be embedded in the daily workload of the implementers. A pre-condition for co-creation is therefore support from all involved stakeholder groups and financial support for all involved parties. Acquiring collaboration partners for the realisation of technical programme aspects (e.g., website building and video development) should be done while considering the unavoidable challenges involved in pursuing the co-creation mindset of continuously adapting intervention aspects. When participating in a co-creation approach, many opinions and suggestions will be put forward. Consequently, managing this enormous and continuous stream of input while continuously balancing it with scientific evidence and theoretical insights is essential in building a representative output. In the co-creation approach, all input must be taken seriously and be weighed according to suitability and feasibility (in terms of technology, time, and finances) to be implemented. The principles of the adaptability of the intervention imply that there cannot be one tightly defined ‘one size fits all’ intervention [13]. Multiple (programme) alternatives (e.g., programme components) should therefore coexist in order to make the programme more compatible with a variety of contexts.

## 4. Conclusions

A co-creation guided approach with continuous micro-process cycles fed by action-oriented research reinforces programme development and implementation by facilitating optimal context-oriented implementation adaptation. Successful co-creation seems to be dependent on how good the intersectoral cooperation between research and practice communities is. A national partner network can provide project-relevant insights and establish capacity building aimed at improving contextual fit. To facilitate such an intersectoral co-creating environment, a time-investment balance needs to be created in and between sectors.

## Figures and Tables

**Figure 1 ijerph-18-07866-f001:**
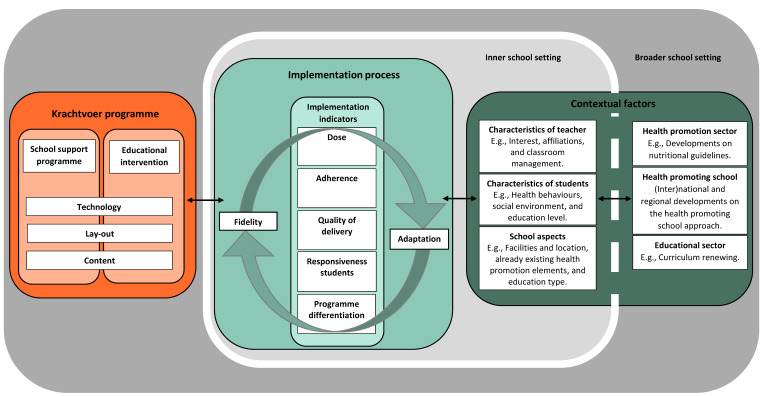
Conceptual framework of contextual fit.

**Table 1 ijerph-18-07866-t001:** Overview of action-oriented research methods and type of data shared in the three programme development phases.

Involved Stakeholders/Participants	Research Methods	Phase and Setting 1: One Partner School 2: Four Partner Schools 3: All Schools Using Krachtvoer	Type of Data Shared
Teachers	Semi-structured interviews	1	Share anticipated programme- and context-related factors influencing programme implementation.
			Share implementation experiences and suggest adjustments to programme components based on implementation experiences.
		2	Share implementation experiences and suggest programme adjustments based on prevailing contextual aspects and implementation experiences.
		3 (Teachers who registered for the evaluation panel)	Share implementation experiences and suggest programme adjustments based on implementation experiences. Share perceived programme- and context-related factors influencing programme implementation.
	Observations	1	Assess the implementation process of programme components and (the interplay of) programme- and context-related factors influencing programme component implementation.
		2	Assess the programme implementation process and (the interplay of) programme- and context-related factors influencing programme implementation.
	Focus group interviews	2	Discuss the programme implementation process and suggest programme adjustments.
	Survey	2	Assess the general characteristics of teachers.
	Evaluation questionnaires on website	3	Assess the characteristics of the teachers who use the programme and the students exposed to the programme, programme appreciation and dose, and suggest programme adjustments.
Students	Semi-structured interviews	1	Share experiences on working with the programme components and suggest adjustments.
		2	Share experiences on working with the programme and suggest adjustments.
	Observations	1	Assess programme component implementation and (the interplay of) programme- and context-related factors influencing programme component implementation.
		2	Assess the programme implementation process and (the interplay of) programme- and context-related factors influencing programme implementation.
	Focus group interviews	2	Share experiences on working with the programme and suggest adjustments.
Parents	Semi-structured interviews	2	Share programme exposure via child and suggest adjustments.
School supporting staff	Semi-structured interviews	2	Share experiences about implementation of the programme in their school setting and relevant context-related factors.
Partner network	Semi-structured interviews	1	Provide input for programme development based on previous experiences and expertise, as well as identified contextual developments within and outside the school setting.
	Partner meetings	All phases	Provide input for the action-oriented research, programme development and adjustments, and sustainable programme use based on own experiences in niche of expertise and fed back action-oriented research results.
Website users	Programme usage monitoring (dashboard)	3	Assess the characteristics of website visitors and how often and which website elements they use.

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
