# Peer review of "Co-Creation Approach with Action-Oriented Research Methods to Strengthen “Krachtvoer”; A School-Based Programme to Enhance Healthy Nutrition in Adolescents"

_ijerph, 2021, doi:10.3390/ijerph18157866_

Round 1

Reviewer 1 Report

First of all, thank you for your work. I do have some major and minor concerns which have to be corrected.

Major corrections:

Introduction

The main aspects of the Krachtvoer program are intake of fruit, snack and drinks, and breakfast/lunch. Please specify in the introduction how the current dietary habits of Dutch adolescents related to these 5 aspects are distinct. You did this for fruit, snacks and drinks (lines # 33-35). Please add further information about Dutch adolescents' breakfast and lunch dietary habits (and add, what exactly is the goal of the Krachtvoer program for these meals).

Page #2, lines #47-48: You state that several interventions have been developed. Please add some references for this statement. Please also add why the Krachtvoer program was developed when there were already many such programs (please highlight the advantages/benefits of the Krachtvoer program).

Page #2, line #50: “…that the conditions for effectiveness (of what? Please specify!) include…”

Page #2, line #83: You state that Krachtvoer is a systematically developed and effective school-based education program. Please add a reference showing that this program is “effective”.

Page #2, lines #88-91: You identified 3 main purposes of this article. However, I had difficulties assigning the manuscript parts properly to each of the 3 themes. If possible, clustering the materials and methods section into 3 parts, with each of them focusing on one of these purposes, would be helpful (for the reader) to follow the text and get answers for all 3 purposes. If this is not possible, think about summarizing the main aspects for each purpose in the discussion.

Methods

Page #3, line #124: Fruit, snacks and drinks, and breakfast and lunch intake have been identified as key determinants. Please add a reference for this statement. Why can these dietary behaviors be influenced by an educational intervention (and other behaviors may not)?

Page #4, line #180: Please introduce the abbreviation “TNO”.

Page #7, line #263: Please add the manufacturer & city of the NVivo software.

Page #3, line #102: You state that there are at minimum 8 lessons about healthy nutrition. I asked myself whether 8 hours are sufficient to achieve a change in dietary knowledge and behavior of the students… Does this also mean that the number of lessons on healthy nutrition performed varies between schools/teachers? How will this be controlled? You may state this aspect in the discussion/limitations.

References

Reference style must be extensively checked and adapted to journal guidelines! Right now, there are frequent mistakes/missing information (e.g., publisher location, different #authors listed, kind of work) present…

Minor corrections:

Page #2, line 58: …seems

Page #8, line #307: “…results reinforces” àresults should be deleted?

Page #8, line #308: There are 2 spaces before “Successful…”

Reviewer 2 Report

Overall Co-creation approach with action-oriented research methods to strengthen “Krachtvoer”; a school-based programme to enhance healthy nutrition in adolescents is a well written and detailed manuscript. 

A few suggestions- May be the authors can include more details such as the time-frame of each phase of the co-creation protocol in the methods. Additionally, details and what kind of interventions should be described in the protocol.

Also, does this protocol account for the number of participants who may not follow all the way through the end ?

What about the situation or if the participants may have some underlying conditions? Does the protocol offers some modifications for them?

Reviewer 3 Report

The manuscript by Driessen-Willems, M. D. et al. entitled “Co-creation approach with action-oriented research methods to strengthen “Krachtvoer”; a school- based programme to enhance healthy nutrition in adolescents” is a study protocol with aim to describe co-creation approach with action-oriented research methods strengthen the Dutch nutrition educational programme for school.

The manuscript is nicely, understandably and systematically written and the topic is very important. Also, it is very important to find an adequate tool to educate vulnerable and rebellious population such as teenagers.

It is going t o be very interesting compare the results of this approach with the past (action-oriented) one.

Some technical comments:

Row 90 – something is missing in the sentence… e.g. …at a young age and are at risk becoming….

Row 169 – CITAVERDE – please provide more details with explanation who are they, what they do, what is their mission, etc.

Question – is there better them for the term “fidelity” from figure 1?

Part of the manuscript from row 241-263 would maybe better fit in the chapter results.

English proofreading is suggested.

Round 2

Reviewer 1 Report

no more comments